# TUBB3 Reverses Resistance to Docetaxel and Cabazitaxel in Prostate Cancer

**DOI:** 10.3390/ijms20163936

**Published:** 2019-08-13

**Authors:** Yohei Sekino, Xiangrui Han, Takafumi Kawaguchi, Takashi Babasaki, Keisuke Goto, Shogo Inoue, Tetsutaro Hayashi, Jun Teishima, Masaki Shiota, Wataru Yasui, Akio Matsubara

**Affiliations:** 1Department of Urology, Graduate School of Biomedical and Health Sciences, Hiroshima University, Hiroshima 734-8551, Japan; 2Department of Molecular Pathology, Graduate School of Biomedical and Health Sciences, Hiroshima University, Hiroshima 734-8551, Japan; 3Department of Urology, Graduate School of Medical Sciences, Kyushu University, Fukuoka 812-8582, Japan

**Keywords:** prostate cancer, TUBB3, PTEN, docetaxel, cabazitaxel, cross-resistance, LY294002

## Abstract

Recent studies have reported that TUBB3 overexpression is involved in docetaxel (DTX) resistance in prostate cancer (PCa). The aim of this study was to clarify the role of TUBB3 in DTX and cabazitaxel (CBZ) resistance, and cross-resistance between DTX and CBZ in PCa. We analyzed the effect of TUBB3 knockdown on DTX and CBZ resistance and examined the interaction between TUBB3 and PTEN. We also investigated the role of phosphoinositide 3-kinases (PI3K) inhibitor (LY294002) in DTX and CBZ resistance. TUBB3 expression was upregulated in DTX-resistant and CBZ-resistant cells. TUBB3 knockdown re-sensitized DTX-resistant cells to DTX and CBZ-resistant cells to CBZ. Additionally, TUBB3 knockdown re-sensitized DTX-resistant cell lines to CBZ, indicating that TUBB3 mediates cross-resistance between DTX and CBZ. Knockdown of TUBB3 enhanced PTEN expression, and PTEN knockout enhanced TUBB3 expression. LY294002 suppressed TUBB3 expression in DTX-resistant and CBZ-resistant cell lines. LY294002 re-sensitized DTX-resistant cell lines to DTX and CBZ-resistant cell lines to CBZ. These results suggest that TUBB3 is involved in DTX resistance and CBZ resistance. A combination of LY294002/DTX and that of LY294002/CBZ could be potential strategies for PCa treatment.

## 1. Introduction

Prostate cancer (PCa) is the most common solid malignancy among men and the second leading cause of cancer-related death in developed countries [1]. Although prostate-specific antigen testing to screen for early-stage PCa is widely used, approximately 30% of patients are newly diagnosed as having locally advanced or metastatic disease [2]. Androgen deprivation therapy is initially effective in treating advanced PCa; however, most of these patients progress to castration-resistant PCa (CRPC) [3]. Docetaxel (DTX) is the standard chemotherapy for CRPC [4], and cabazitaxel (CBZ) is the next-generation taxane drug for CRPC after DTX treatment [5]. Despite improvements in outcomes, nearly all patients treated with DTX and CBZ become refractory due to the development of resistance to these therapies. Additionally, several recent reports have shown cross-resistance between DTX and CBZ [6,7]. Unfortunately, until now, there has been no established marker for DTX and CBZ treatment. Therefore, clarifying the mechanisms of DTX and CBZ resistance and cross-resistance between DTX and CBZ will greatly improve outcomes for patients with CRPC.

Taxanes block cell mitosis and induce apoptosis in tumor cells by targeting microtubule activity [8]. Microtubules are composed of polymers of α- and β-tubulin heterodimers [9]. βIII-tubulin encoded by the *TUBB3* gene is a microtubule protein normally expressed in neuronal cells. A recent study has shown TUBB3 to be associated with the progression of CRPC [10]. What is more, some studies have reported that overexpression of TUBB3 is involved in DTX resistance in some cancers including PCa [11,12,13]. The status of TUBB3 expression has predictive value in terms of overall survival in patients treated with DTX in PCa [14]. Another report has shown that TUBB3 is associated with CBZ resistance in breast cancer [15,16]. Collectively, these results suggest that TUBB3 is involved in PCa progression and plays an important role in resistance to taxanes across cancer types. However, the exact mechanism by which TUBB3 mediates taxane resistance in PCa has not been fully elucidated.

In this study, we use DTX- and CBZ-resistant PCa cell lines to analyze the involvement of TUBB3 in, and the effect of TUBB3 knockdown on, DTX and CBZ resistance in PCa cell lines. We also investigate the effect of LY294002 in these DTX- and CBZ-resistant PCa cell lines.

## 2. Results

### 2.1. Characterization of DTX- and CBZ-Resistant PCa Cell Lines

We used two DTX-resistant (DU145-DR, C4-2-DR) [17,18,19] and two CBZ-resistant PCa cell lines (LNCaP-CR, 22Rv-1-CR) [20] to analyze the involvement of TUBB3 in DTX and CBZ resistance. We performed 4,5-dimethylthiazol-2-yl-2,5-diphenyltetrazolium bromide (MTT) assays to measure cell viability under various concentrations of CBZ in parental and CBZ-resistant cells in LNCaP and 22Rv-1 cells. The IC50 values of LNCaP-CR and 22Rv-1-CR cells were significantly higher than those of LNCaP and 22Rv-1 parental cells (Figure 1A). We compared the expression of c-PARP, which was used as an apoptosis marker, in parental and CBZ-resistant cells in LNCaP and 22Rv-1 cells. Western blotting showed that the expression of c-PARP was induced by CBZ treatment in LNCaP and 22Rv-1 parental cells but was not changed by CBZ treatment in LNCaP-CR and 22Rv-1-CR cells (Figure 1B). Recent studies have reported cross-resistance between DTX and CBZ in PCa [6,7]. MTT assays were performed to measure cell viability under various concentrations of CBZ in DTX-resistant cells and parental cells in DU145 and C4-2 cell lines. The IC50 values of DTX-resistant cells were significantly higher than those of parental cells in DU145 and C4-2 cells (Figure 1C). Furthermore, MTT assays were performed to measure cell viability under various concentrations of DTX in CBZ-resistant cells and parental cells in LNCaP and 22Rv-1 cell lines. The IC50 values of CBZ-resistant cells were significantly higher than those of parental cells in LNCaP and 22Rv-1 cells (Figure 1D). These results indicate that cross-resistance between DTX and CBZ exists in PCa.

### 2.2. TUBB3 is Overexpressed in DTX- and CBZ-Resistant PCa Cell Lines

Given the previous reports that TUBB3 is associated with cancer progression and DTX resistance in PCa [10,14], we sought to examine the expression of TUBB3 in a normal prostate cell line (RWPE-1) and PCa cell lines. Western blotting showed that the expression of TUBB3 was not detected in RWPE-1 cells but was detected in the PCa cell lines. To verify whether TUBB3 is involved in DTX and CBZ resistance, we investigated the expression of TUBB3 in two DTX-resistant PCa cell lines and two CBZ-resistant PCa cell lines. Western blotting and a quantitative reverse transcriptase polymerase chain reaction (qRT-PCR) showed that the expression of TUBB3 was overexpressed in DTX-resistant cells compared with parental DU145 and C4-2 cells at both mRNA and protein levels (Figure 2B, Appendix A), which was consistent with findings of a previous study [12]. Furthermore, Western blotting and a qRT-PCR showed that TUBB3 was overexpressed in LNCaP-CR and 22Rv-1-CR cells compared with parental LNCaP and 22Rv-1 cells (Figure 2B, Appendix A), which is the first evidence of this to the best of our knowledge.

### 2.3. Inhibition of TUBB3 Reverses DTX and CBZ Resistance

We next analyzed whether knockdown of TUBB3 improves DTX and CBZ sensitivity in DTX- and CBZ-resistant PCa cells. We measured cell viability in DU145-DR and C4-2-DR cells with knockdown of TUBB3 under various concentrations of DTX. Western blotting showed that the expression of TUBB3 was suppressed in DU145-DR and C4-2-DR cells transfected with small interfering RNAs (siRNAs) for TUBB3 compared with that in DU145-DR and C4-2-DR cells transfected with siRNAs for negative control (Figure 3A). We found that downregulation of TUBB3 caused DU145-DR and C4-2-DR cells to become re-sensitized to DTX treatment (Figure 3B). Additionally, we measured cell viability in LNCaP-CR and 22Rv-1-CR cells with knockdown of TUBB3 under various concentrations of CBZ. Western blotting showed that the expression of TUBB3 was suppressed in LNCaP-CR and 22Rv-1-CR cells transfected with siRNAs for TUBB3 compared with that in LNCaP-CR and 22Rv-1-CR cells transfected with siRNAs for the negative control (Figure 3C). We observed that downregulation of TUBB3 re-sensitized the LNCaP-CR and 22Rv-1-CR cells to CBZ treatment (Figure 3D). Furthermore, we determined whether increased expression of TUBB3 is responsible for cross-resistance between DTX and CBZ. We measured cell viability in DU145-DR and C4-2-DR cells with knockdown of TUBB3 under various concentrations of CBZ. As we expected, downregulation of TUBB3 re-sensitized DTX-resistant cells to CBZ treatment (Figure 3E).

### 2.4. Interaction Between PTEN and TUBB3 Expression

Several recent reports have shown an interaction between the expression of PTEN and that of TUBB3 in some cancers including PCa [21,22]. To examine the interaction between PTEN and TUBB3 in PCa, we analyzed the expression of *PTEN* and *TUBB3* in 28 PCa tissues by qRT-PCR. There was a significant inverse correlation between *PTEN* and *TUBB3* (*p* < 0.001, R = –0.64) (Figure 4A). In this sample set, the expression of *TUBB3* was significantly associated with a high Gleason score (*p* = 0.048) and high tumor stage (*p* < 0.001) (Figure 4B). We also investigated the effect of TUBB3 knockdown on the expression of PTEN in DU145 and 22Rv-1 cells, which are PTEN wild-type PCa cell lines [23]. Western blotting showed that knockdown of TUBB3 enhanced the expression of PTEN in the DU145 and 22Rv-1 cells (Figure 4C). Further, to examine the effect of PTEN modulation on TUBB3 expression, we generated PTEN knockout cells by using a PTEN-CRISPR vector in DU145 and 22Rv-1 cells. Western blotting showed that PTEN expression was not detected and p-Akt expression was upregulated in DU145 and 22Rv-1 cells transfected with the PTEN-CRISPR vector compared with that in DU145 and 22Rv-1 cells transfected with an empty vector (Figure 4D). PTEN knockout enhanced the expression of TUBB3 compared with an empty vector in DU145 and 22Rv-1 cells (Figure 4E). These results indicate a reciprocal interaction between PTEN and TUBB3 in PCa.

### 2.5. LY294002 Improves Taxane Resistance

It is often reported that PTEN loss or activation of the phosphoinositide 3-kinases (PI3K)/AKT pathway leads to progression of CRPC [24,25]. Therefore PI3K/AKT signaling has attracted attention as a therapeutic target [26]. LY294002 is a strong inhibitor of PI3K/AKT [27]. Recent studies showed that LY294002 suppressed the expression of TUBB3 in colon cancer and lung cancer [28,29]. We therefore investigated the expression of TUBB3 under LY294002 treatment in C4-2-DR and LNCaP-CR cells, which are PTEN-null PCa cells [30]. Western blotting showed that LY294002 reduced the expression of TUBB3 in C4-2-DR and LNCaP-CR cells (Figure 5A). A recent study has shown that LY294002 improved DTX sensitivity in LNCaP cells [31]. Thus, we investigated the effect of combination therapy with DTX and LY294002 in C4-2-DR cells and combination therapy with CBZ and LY294002 in LNCaP-CR cells. We measured cell viability under DTX alone or in combination with DTX and LY294002 in parental and DTX-resistant C4-2 cells. DTX alone had little effect on cell viability in C4-2-DR cells. However, the combination of DTX and LY294002 significantly reduced cell viability in the C4-2-DR cells (Figure 5B). Western blotting revealed that the expression of cleaved PARP (c-PARP) was induced by combination therapy with DTX and LY294002 compared to LY294002 treatment alone in C4-2-DR cells (Figure 5C). We measured cell viability under CBZ alone or in combination with LY294002 in parental and CBZ-resistant LNCaP cells. CBZ alone had little effect on cell viability in LNCaP-CR cells, whereas the combination of CBZ and LY294002 significantly reduced cell viability in LNCaP-CR cells (Figure 5D). Western blotting revealed that the expression of c-PARP was induced by combination therapy with CBZ and LY294002 compared to LY294002 treatment alone in LNCaP-CR cells (Figure 5E).

## 3. Discussion

DTX is the first-line therapy for patients with metastatic CRPC. Recent clinical studies have shown that early DTX treatment with androgen deprivation therapy leads to better overall survival in comparison to androgen deprivation therapy alone in patients with metastatic hormone-sensitive PCa [32,33]. CBZ is approved only for those patients who have previously undergone DTX treatment [5]. A recent clinical trial showed that CBZ treatment improved overall survival after DTX treatment. These findings indicate that DTX and CBZ are becoming increasingly important in hormone-sensitive PCa and CRPC therapy. However, disease relapse eventually occurs due to the development of resistance to DTX or CBZ. In the present study, we showed that the expression of TUBB3 was increased in both DTX- and CBZ-resistant cells. Knockdown of TUBB3 improved sensitivity to DTX in DTX-resistant PCa cells and to CBZ in CBZ-resistant PCa cells, indicating that TUBB3 may be involved in a common resistance mechanism between DTX and CBZ in PCa. Several studies have reported that the expression of TUBB3 was higher in the PCa region than that in the non-neoplastic region [10,14]. Western blotting also showed that the expression of TUBB3 was not detected in RWPE-1 cells. Collectively, these results suggest that the expression of TUBB3 is specific to PCa, and TUBB3 may be a promising therapeutic target to overcome taxane resistance in PCa.

Recently, several studies reported that cross-resistance exists between DTX and CBZ treatment in PCa cell lines [6,7,34]. In the present study, we found DTX-resistant cells to be more resistant to CBZ treatment than parental cells in DU145 and C4-2 cells, and CBZ-resistant cells to be more resistant to DTX treatment than parental cells in LNCaP and 22RV-1 cells. These results indicate the possibility of cross-resistance between DTX and CBZ. Taxane cross-resistance has not been well studied in clinical trials. The TROPIC clinical trial showed that the median overall survival of patients with metastatic CRPC treated with CBZ after DTX was 15.1 months [5], whereas that in the FIRSTANA clinical trial of patients with metastatic CRPC treated with CBZ in the first-line setting was 25.2 months. Although further clinical studies are needed to verify our current findings, they suggest that the response to CBZ may be diminished after DTX partly due to cross-resistance.

PTEN loss is an early and stable event in the carcinogenesis process and is associated with poor prognosis in PCa. The expression of PTEN is regulated by mutations, epigenetic and transcriptional silencing, post-translational regulation, and protein–protein interactions [35]. Some studies have shown that the expression of TUBB3 was inversely correlated with the expression of PTEN [21,22]. In the present study, Western blotting showed that knockdown of TUBB3 induced the expression of PTEN. Additionally, knockout of PTEN induced the expression of TUBB3. We also showed that the expression of TUBB3 was inversely correlated with the expression of PTEN in PCa tissues by qRT-PCR. These results indicate that there is a reciprocal interaction between TUBB3 and PTEN in PCa. A recent study has shown that PTEN loss was an independent prognostic factor in DTX treatment [36]. PTEN acts as a phosphatase regulator of the PI3K/AKT pathway, which is also involved in DTX resistance [37]. Although further studies are needed to elucidate the mechanism of interaction between TUBB3 and PTEN, these findings indicate a potential mechanistic explanation for TUBB3 restoring sensitivity to DTX and CBZ in PCa.

LY294002 is a synthetic compound that was designed as a PI3K inhibitor [27]. Several studies have reported the potential role of LY294002 in PCa [38,39]. A recent report showed the usefulness of combination therapy with DTX and LY294002 in LNCaP cells [31]. In the present study, we showed that LY294002 re-sensitized DTX-resistant cells to DTX. Additionally, LY294002 also re-sensitized CBZ-resistant cells to CBZ, which we believe is the first evidence of this occurrence. However, the exact mechanism for LY294002 improvement of the sensitivity to DTX and CBZ is not well understood. LY294002 has been shown to inhibit several key signals, such as NF-κB, AKT phosphorylation and survivin in addition to PI3K [40]. In the present study, LY294002 suppressed the expression of TUBB3 in C4-2 DR and LNCaP CR cells, which may help to explain the restoration of the sensitivity to DTX and CBZ by LY294002. In current cancer treatments, different types of chemotherapeutic agents are combined to improve efficacy and to reduce toxicity. Collectively, these results imply that combination therapy of LY294002 with DTX and that of LY294002 with CBZ may be promising strategies to overcome DTX and CBZ resistance in PCa.

There are some limitations in this study. We performed qRT-PCR analysis for PTEN and TUBB3 expression in a relatively small sample size, and the samples were not derived from CRPC patients. Therefore, a study with a larger number of CRPC patients will be necessary to verify the current findings. Second, we showed that the combination of LY294002 with DTX or CBZ had higher efficiency than DTX or CBZ alone chemotherapy. However, we did not clarify whether the combination is synergistic or additive. Third, so far, several PI3K inhibitors have been reported, and several clinical trials using PI3K inhibitors are ongoing [41]. Further studies using other PI3K inhibitors against DTX and CBZ resistance could support the potential of PI3K inhibitors in the treatment of PCa. 

In summary, we showed that the expression of TUBB3 was upregulated in DTX- and CBZ-resistant cells in PCa. TUBB3 was involved in both DTX resistance and CBZ resistance. There was a reciprocal interaction between TUBB3 and PTEN in PCa. TUBB3 mediated cross-resistance between DTX and CBZ. LY294002 re-sensitized cells resistant to DTX treatment and also re-sensitized cells resistant to CBZ treatment. The data presented here highlight the great potential of combination therapy of LY294002 with DTX and that of LY294002 with CBZ in chemotherapy for PCa.

## 4. Materials and Methods

### 4.1. Cell Lines

Four PCa cell lines (LNCaP, DU145, 22Rv-1, and C4-2), two DTX-resistant PCa cell lines (DU145-DR, C4-2-DR) and two CBZ-resistant PCa cell lines (LNCaP-CR, 22Rv-1-CR) were kindly provided by Masaki Shiota (Kyushu University, Fukuoka, Japan). DU145 cells were maintained in MEM (Nissui Pharmaceutical Co. Ltd., Tokyo, Japan). LNCaP, C4-2, and 22Rv-1 cells were maintained in RPMI 1640 (Nissui Pharmaceutical Co. Ltd.) containing 10% fetal bovine serum (BioWhittaker, Walkersville, MD, USA), 2 mM L-glutamine, 50 U/mL penicillin, and 50 g/mL streptomycin in a humidified atmosphere of 5% CO_2_ and 95% air at 37 °C. DTX-resistant PCa cell lines were cultured under DTX (DU145: 2 ng/mL and C4-2-DR: 5 ng/mL). CBZ-resistant cell lines were cultured under CBZ (LNCaP-CR: 2 ng/mL and 22Rv-1-CR: 5 ng/mL).

### 4.2. DTX, CBZ, and LY294002 Treatment

DTX, CBZ, and LY294002 were obtained from Funakoshi (Tokyo, Japan). An MTT assay was performed to analyze drug sensitivities as described previously [19]. At 24 h after transfection of siRNAs for TUBB3 or negative control, these cells were exposed to DTX or CBZ for 48 h. For combination therapy, an MTT assay was performed after combination with DTX/LY294002 or CBZ/LY294002. Drug sensitivity curves and IC50 values were calculated using GraphPad Prism 4.0 software (GraphPad Software, La Jolla, CA, USA) [42].

### 4.3. Western Blotting Analysis

For Western blotting analysis, cells were lysed as described previously [43]. Primary antibody, TUBB3 (BioLegend, San Diego, CA, USA), PTEN, Akt, phosphorylation Akt (p-Akt), and cleaved PARP (Cell Signaling Technology, Inc., Danvers, MA, USA) were used. β-Actin (Sigma-Aldrich, St. Louis, MO, USA) was used as a loading control. The ID and dilution of primary and secondary antibodies are summarized in Table 1.

### 4.4. qRT-PCR Analysis

Extraction of total RNA, synthesis of cDNA, and qRT-PCR were performed as described previously [44]. Total RNA was isolated from frozen samples or cell lines by NucleoSpin RNA (Takara Bio, Shiga, Japan). The quality and concentration of RNA was determined by an ultraviolet spectrophotometer: Nano Drop (Thermo Fisher Scientific, Waltham, MA, USA). A total of 1 µg of total RNA was converted to complementary DNA with a first-strand complementary DNA synthesis kit (Amersham Biosciences, Piscataway, NJ, USA). Real-time detection of the emission intensity of SYBR Green bound to double-stranded DNA was performed with a CFX Connect real-time system (Bio-Rad Laboratories, Hercules, CA, USA). ACTB-specific PCR products, which were amplified from the same RNA samples, served as internal controls. The primer sequences are summarized in Table 2.

### 4.5. Tissue Samples

We obtained 28 PCa tissue samples for qRT-PCR. The clinicopathologic features are summarized in Appendix A. The samples were immediately frozen in liquid nitrogen and stored at −80 °C until use. Twenty-eight samples were collected from patients at Hiroshima University Hospital and written comprehensive approvals for basic or clinical research were obtained from each patient. This study was conducted in accordance with the Ethical Guidance for Human Genome/Gene Research of the Japanese Government. The Institutional Review Board of Hiroshima University Hospital approved this study (approval no. E-688; September 7, 2017).

### 4.6. RNA Interference

Silencer® Select (Ambion, Austin, TX, USA) against TUBB3 was used for RNA interference. Two independent oligonucleotides and negative control siRNA (Invitrogen, Carlsbad, CA, USA) were used [42].

### 4.7. Generation of PTEN Knockout Cells

To knock out PTEN in DU145 and 22Rv-1 cells, we used CRISPR-Cas9 technology, which was performed as described previously [45]. PTEN single-guide RNAs (PTEN-CRISPR vector) or scrambled single-guide RNAs (empty vector), and CRISPR/Cas9 All-in-One lentivector pLenti-U6-sgRNA-SFFV-Cas9-2A-Puro were purchased from ABM Inc. (Richmond, BC, Canada). The two sgRNA sequences of PTEN-CRISPR vector were TGGGAATAGTTACTCCC (#1) and CTTGTCTTCCCGTCGTG (#2). Lentiviral particles were generated by co-transfection of HEK293T cells with Cas9-sgRNA constructs and packaging plasmids (GAG, VSVG, REV). After 48 h, the conditioned media containing lentiviral particles were harvested and used to infect cells using polybrene as the transfection agent. Stable PTEN knockout cells were selected by passaging in media containing 4 µg/mL puromycin.

### 4.8. Statistical Analysis

Statistical differences were evaluated using the two-tailed Student’s *t*-test or Mann–Whitney U test. A *p*-value of <0.05 was considered statistically significant. Statistical analyses were conducted primarily using GraphPad Prism software (GraphPad Software Inc., San Diego, CA, USA).

## Figures and Tables

**Figure 1 ijms-20-03936-f001:**
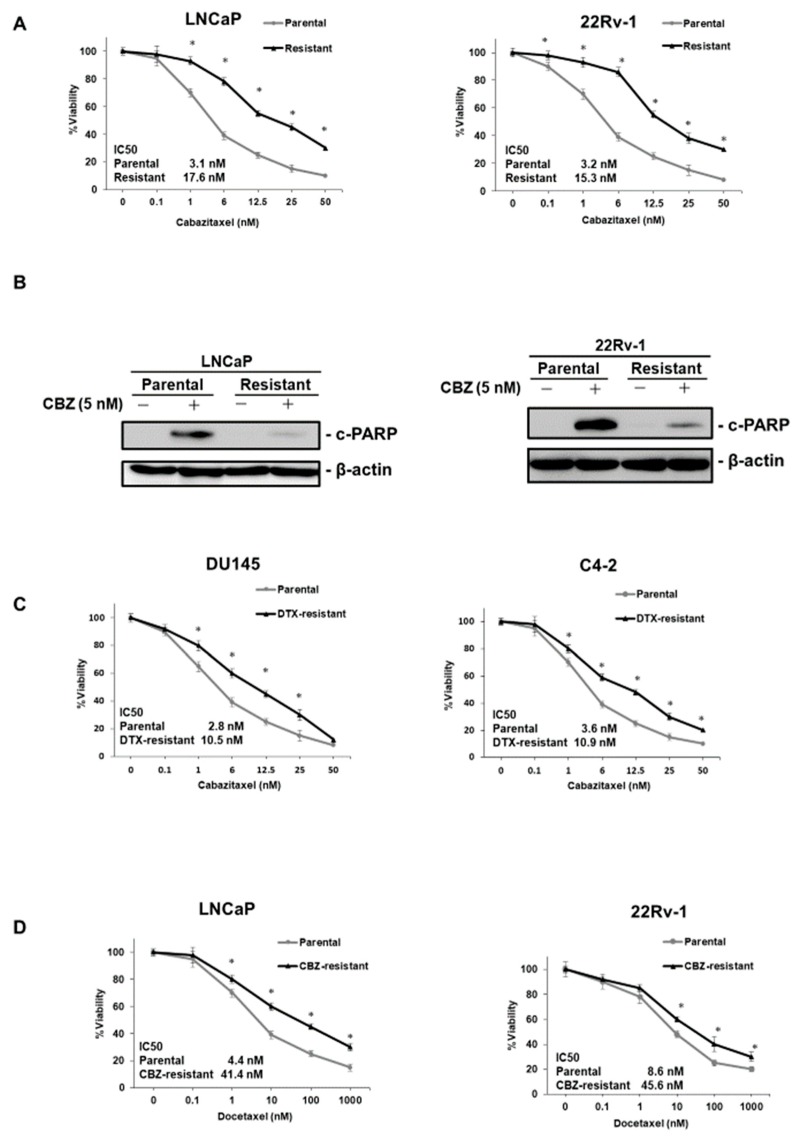
Characterization of cabazitaxel (CBZ)-resistant prostate cancer cell lines. (**A**) The dose-dependent effects of CBZ on the cell viability of parental and CBZ-resistant cell lines in LNCaP and 22Rv-1 cells. The results are expressed as the mean and standard deviation (SD) of triplicate measurements. * *p* < 0.01. (**B**) Western blotting of c-PARP in parental and CBZ-resistant cell lines in LNCaP and 22Rv-1 cells in the presence of CBZ (5 nM) or vehicle (ethanol). β-actin was used as a loading control. c-PARP: cleaved PARP. (**C**) The dose-dependent effects of CBZ on the cell viability of parental and docetaxel (DTX)-resistant cell lines in DU145 and C4-2 cells. The results are expressed as the mean and SD of triplicate measurements. * *p* < 0.01. (**D**) The dose-dependent effects of DTX on the cell viability of parental and CBZ-resistant cell lines in LNCaP and 22Rv-1 cells. The results are expressed as the mean and SD of triplicate measurements. * *p* < 0.01.

**Figure 2 ijms-20-03936-f002:**
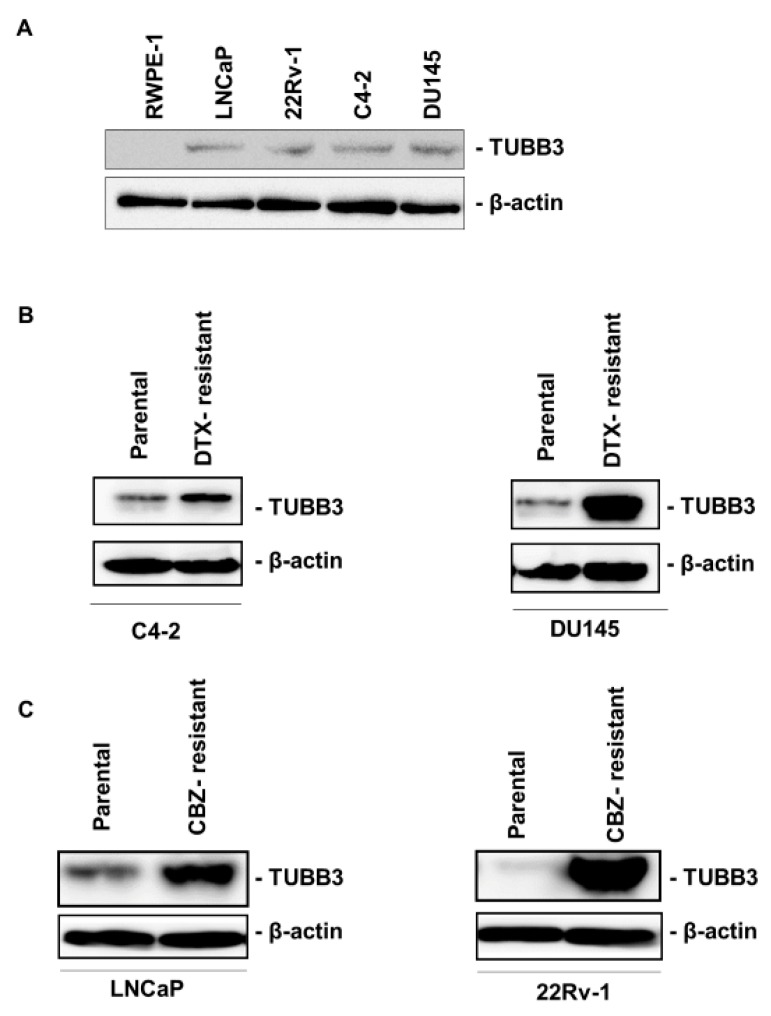
TUBB3 is overexpressed in docetaxel (DTX)-resistant cell lines and cabazitaxel (CBZ)-resistant cell lines. (**A**) Western blotting of TUBB3 in RWPE-1, LNCaP, 22Rv-1, C4-2 and DU145 cells. β-actin was used as a loading control. (**B**) Western blotting of TUBB3 in parental and DTX-resistant cell lines in C4-2 and DU145 cells. β-actin was used as a loading control. (**C**) Western blotting of TUBB3 in parental and CBZ-resistant cell lines in LNCaP and 22Rv-1 cells. β-actin was used as a loading control.

**Figure 3 ijms-20-03936-f003:**
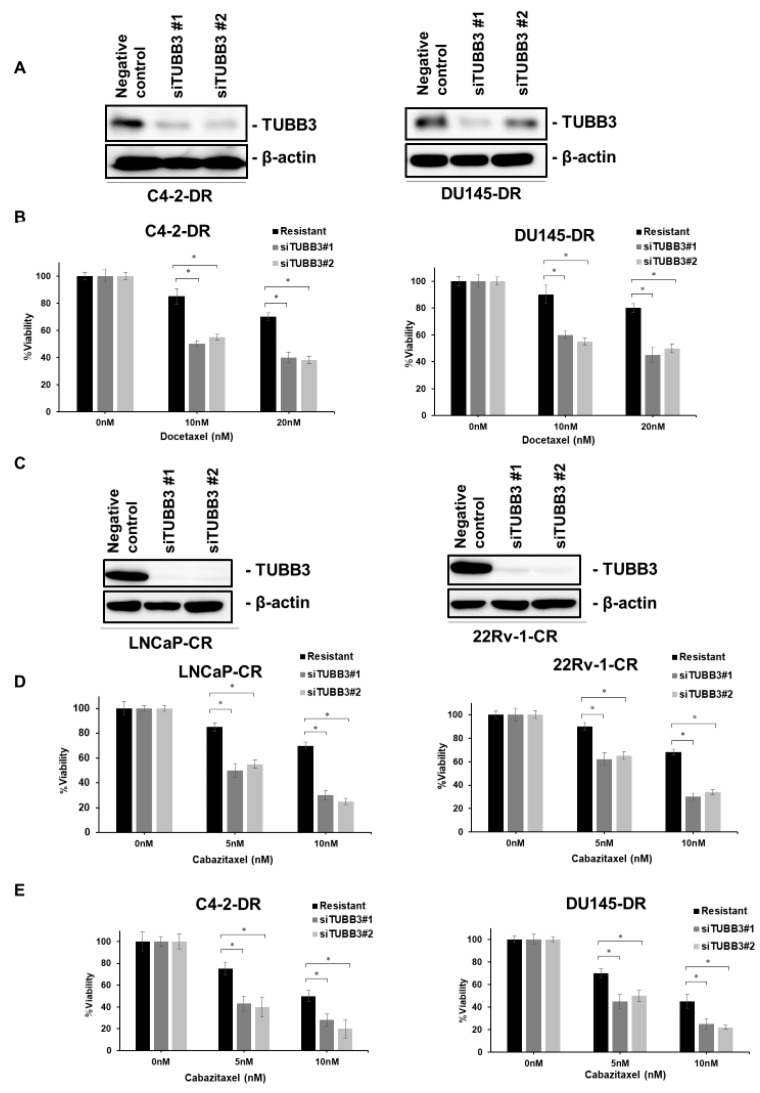
Inhibition of TUBB3 reverses docetaxel (DTX) resistance and cabazitaxel (CBZ) resistance in vitro. (**A**) Western blotting of TUBB3 in C4-2-DR and DU145-DR cells transfected with a negative control or two different small interfering RNAs (siRNAs) for TUBB3. β-actin was used as a loading control. (**B**) The effects of DTX (10, 20 nM) on the cell viability of C4-2-DR and DU145-DR cells transfected with a negative control or two different siRNAs for TUBB3. The results are expressed as the mean and SD of triplicate measurements. * *p* < 0.01. (**C**) Western blotting of TUBB3 in LNCaP-CR and 22Rv-1-CR cells transfected with a negative control or two different siRNAs for TUBB3. β-actin was used as a loading control. (**D**) The effects of CBZ (5, 10 nM) on the cell viability of LNCaP-CR and 22Rv-1-CR cells transfected with a negative control or two different siRNAs for TUBB3. The results are expressed as the mean and SD of triplicate measurements. * *p* < 0.01. (**E**) The effects of CBZ (5, 10 nM) on the cell viability of C4-2-DR and DU145-DR cells transfected with a negative control or two different siRNAs for TUBB3. The results are expressed as the mean and SD of triplicate measurements. * *p* < 0.01.

**Figure 4 ijms-20-03936-f004:**
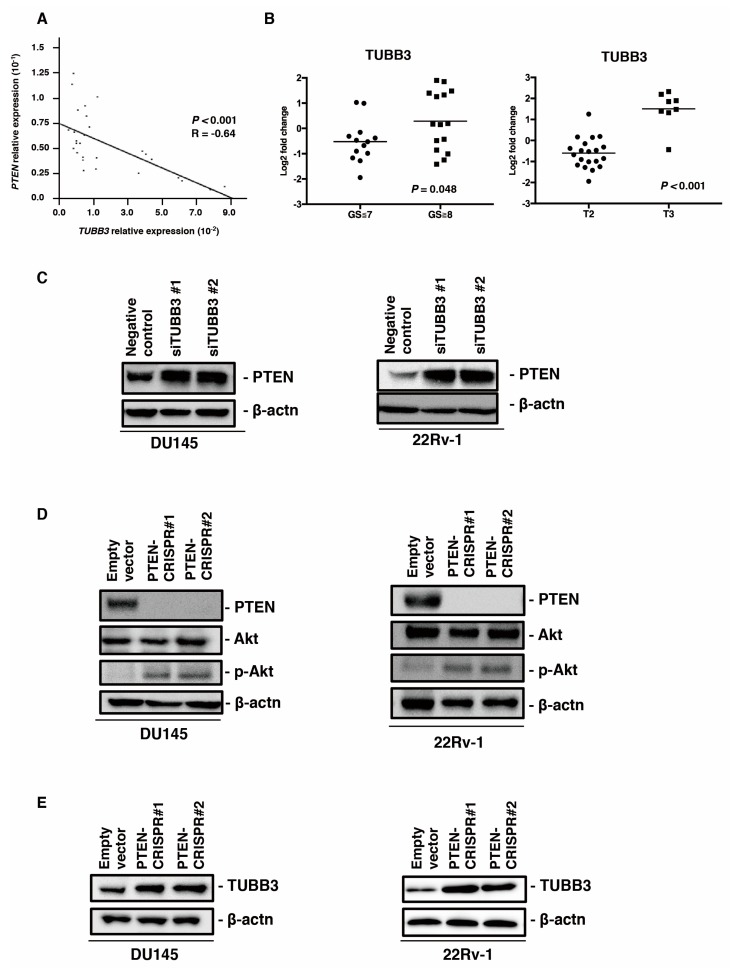
The interaction between TUBB3 and PTEN in prostate cancer (PCa). (**A**) The correlation between TUBB3 and PTEN in PCa tissues. Spearman’s correlation coefficient and *p*-values are indicated. (**B**) Scatter plot diagrams showing the association between the expression of TUBB3 and clinicopathological findings (Gleason score (GS), tumor stage (T)). *p*-values are indicated. (**C**) Western blotting of PTEN in DU145 and 22Rv-1 transfected with negative control or two different siRNAs for TUBB3. β-actin was used as a loading control. (**D**) Western blotting of PTEN, Akt, and phosphorylation Akt (p-Akt) in DU145 and 22Rv-1 cells transfected with an empty vector or two different PTEN-CRISPR vectors. β-actin was used as a loading control. (**E**) Western blot analysis of TUBB3 in DU145 and 22Rv-1 cells transfected with an empty vector or two different PTEN-CRISPR vectors. β-actin was used as a loading control.

**Figure 5 ijms-20-03936-f005:**
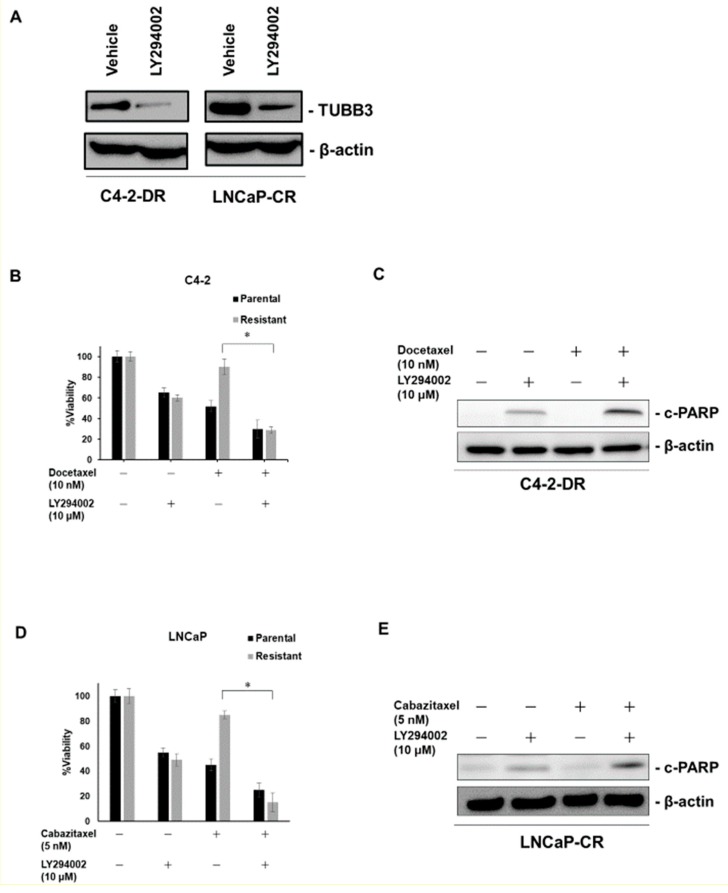
The effect of phosphoinositide 3-kinases (PI3K) inhibitor LY294002 on docetaxel (DTX) and cabazitaxel (CBZ) resistance. (**A**) Western blotting of TUBB3 in C4-2 DTX-resistant (C4-2-DR) cells and LNCaP CBZ-resistant (LNCaP-CR) cells treated with vehicle (ethanol) and LY294002 (20 µM). β-actin was used as a loading control. (**B**) The effect of combination therapy with DTX (10 nM) and LY294002 (10 µM) on cell viability in parental and DTX-resistant cell lines in C4-2 cells. The results are expressed as the mean and SD of triplicate measurements. * *p* < 0.01. (**C**) Western blotting of c-PARP in DTX-resistant cell lines in C4-2 cells treated with either LY294002 (10 nM) alone or with DTX (10 nM). β-actin was used as a loading control. c-PARP: cleaved PARP. (**D**) The effect of combination therapy with DTX (10 nM) and LY294002 (10 µM) on cell viability in parental and CBZ-resistant cell lines in LNCaP cells. The results are expressed as the mean and SD of triplicate measurements. * *p* < 0.01. (**E**) Western blotting of c-PARP in CBZ-resistant cell lines in LNCaP cells treated with either LY294002 (10 nM) alone or with CBZ (5 nM). β-actin was used as a loading control.

**Table 1 ijms-20-03936-t001:** ID and dilution of primary and secondary antibodies.

Antibody	ID	Dilution
Primary antibody		
TUBB3	MMS-435P	1:500
PTEN	138G6	1:500
Akt	9272	1:500
p-Akt	9271	1:500
Cleaved PARP	5625	1:500
β-actin	A5441	1:10,000
Secondary antibody		
Anti-IgG (H+L chain) (Mouse) pAb-HRP	330	1:500
Anti-IgG (H+L chain) (Rabbit) pAb-HRP	458	1:500

**Table 2 ijms-20-03936-t002:** Primers sequences for quantitative reverse transcriptase polymerase chain reaction (qRT-PCR).

Gene	Forward Primer	Reverse Primer
*TUBB3*	GAGATGGAGTTCACCGAGGC	TCGTCTTCGTACATCTCGCC
*PTEN*	ACCCACCACAGCTAGAACTT	GGGAATAGTTACTCCCTTTTTGTC
*ACTB*	TCACCGAGCGCGGCT	TAATGTCACGCACGATTTCCC

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
