# Peer review of "TUBB3 Reverses Resistance to Docetaxel and Cabazitaxel in Prostate Cancer"

_ijms, 2019, doi:10.3390/ijms20163936_

Round 1

Reviewer 1 Report

This study was aimed to the analysis of the role of TUBB3 in Docetaxel and Cabazitaxel resistance in prostate cancer. Authors found that TUBB3 is involved in both Docetaxel and also Cabazitaxel resistance in prostate cancer cell line models. In PCa tumor samples, interaction between TUBB3 and PARP expression was found. Finally, PI3K inhibitor (LY294002) re-sensitized cells resistant to Docetaxel and also Cabazitaxel and combination of this inhibitor with Docetaxel or Cabazitaxel may have potential for higher efficiency of prostate cancer chemotherapy. Subscribed manuscript is capable of being published after a minor revision mainly in the section, where PCa tumor samples from patients were used.

Major points

Methods; 4.4. Although the authors mentioned that the extraction of RNA, preparation of cDNA and qRT-PCR was performed as described in previous study, they should mention, which type of analysis of qRT-PCR expression data was used.

Methods; 5. There is not declared, how exactly was isolated RNA from 28 PCa tissue samples and how was RNA quantity and quality estimated from tumor tissue source of samples. Did the authors estimate RIN of RNA extracted from PCa tumor tissue samples?

Supplementary Materials; Are there available any other clinical and pathological features such as Stage, Grade and Survival of PCa patients involved in the study? It could be interesting to analyse associations of TUBB3 expression levels with clinical and pathological data and also survival of PCa patients.

Author Response

Our responses to the reviewers' comments for ijms-567404

Title:TUBB3 reverses resistance to docetaxel and cabazitaxel in prostate cancer

Authors:Yohei Sekino * , Xiangrui Han , Takafumi Kawaguchi , Takashi Babasaki , Keisuke Goto , Shogo Inoue , Tetsutaro Hayashi , Jun Teishima , Masaki Shiota , Wataru Yasui , Akio Matsubara

Associate Editor: We would like to express our deepest gratitude to the associate editor for all comments. According to the associate editor's and reviewer's comments, we have thoroughly revised the manuscript.

Reviewers:We thank the reviewers for the critical comments which have helped us to improve the manuscript. We have addressed all criticisms as follows.

Reviewer #1

Q1. The reviewer required us to describe the detail of qRT-PCR in the method section

We thank the reviewer for the valuable suggestion. We added the description of qRT-PCR in the method section.

These statements were described in the revised method (page 14, line 290-296).

Q2. The reviewer proposed that we should mention the quality of 28 PCa tissue samples.

We thank the reviewer for the important comment. Unfortunately, we did not analyze the RIN score of RNA extracted from PCa tissue samples. In this study, we used Nano Drop spectrophotometer to determine the quality of RNA from samples.

These statements were described in the revised method (page 14, line 291-292).

Q3, 4. The reviewer suggested that we should study the association TUBB3 expression with clinicopathological factors.

We are grateful for the reviewer’s important comment. We added the description about association between TUBB3 and clinicopathological factors. Unfortunately, we do not have the survival date of 28 PCa tissue samples.

These statements were described in the revised results (page 8, line 139-141).

Reviewer 2 Report

In the present manuscript Sekino and co-workers reported the role of TUBB3 in docetaxel and cabazitaxel resistance in prostate cancer cells. The authors demonstrated that the KO of TUBB3 in resistant cells resulted in sensitivity to docetaxel and cabazitaxel, and that resistant cells are sensitive to PI3K inhibitor LY294002, which produces down regulation of TUBB3. Finally, the authors suggest a combination therapy docetaxel/LY294002 as well as cabazitaxel/LY294002. The paper is well written and suffers from minor imperfections:

1-Regarding LY294002, authors used concentration of 10 μM. Which is the effect of these concentration exposure on cell lines considered. Authors are requested to perform dose dependent cytotoxic experiments.  

3-Figure 4B. Please note the discrepancy between image and figure legend. Authors are requested to clarify which cell lines were used (sensitive or resistant).

Author Response

Our responses to the reviewers' comments for ijms-567404

Title:TUBB3 reverses resistance to docetaxel and cabazitaxel in prostate cancer

Authors:Yohei Sekino * , Xiangrui Han , Takafumi Kawaguchi , Takashi Babasaki , Keisuke Goto , Shogo Inoue , Tetsutaro Hayashi , Jun Teishima , Masaki Shiota , Wataru Yasui , Akio Matsubara

Associate Editor: We would like to express our deepest gratitude to the associate editor for all comments. According to the associate editor's and reviewer's comments, we have thoroughly revised the manuscript.

Reviewers:We thank the reviewers for the critical comments which have helped us to improve the manuscript. We have addressed all criticisms as follows.

Reviewer #2

Q1. The reviewer requested us to perform dose dependent cytotoxic experiments using LY294002.

We totally agree with the reviewer’s opinion. Because 10µM concentration of LY294002 suppressed nearly 50% cell viability as shown in figure 4B and 4C, we decided to use this concentration. In the future, we will surely analyze dose dependent cytotoxic experiments to clarify whether the combination is synergistic or additive. We added the description to the limitation in the discussion section.

These statements were described in the revised discussion (page 13, line 250-253).

Q2. The reviewer requested us to modify the figure legend in figure 4.

We thank the reviewer for the valuable comment. We are terribly sorry for a silly mistake. We have closely reviewed and revised it.

These statements were described in the revised results (page 9, line 155-156).